# Fibroblast Growth Factor-23 (FGF-23) in Dogs—Reference Interval and Correlation with Hematological and Biochemical Parameters

**DOI:** 10.3390/ani13203202

**Published:** 2023-10-13

**Authors:** Sandra Lapsina, Nicole Nagler, Simon Franz Müller, Annette Holtdirk, Tanja Kottmann, Elisabeth Müller, Ingo Schäfer

**Affiliations:** 1LABOKLIN GmbH and Co. KG, Steubenstraße 4, 97688 Bad Kissingen, Germany; lapsina@laboklin.com (S.L.);; 2Dr. MED. Kottmann—Clinical Research Organization, Beverstraße 64, 59077 Hamm, Germany

**Keywords:** canine, chronic kidney disease, renal, clinical pathology, FGF-23

## Abstract

**Simple Summary:**

Fibroblast growth factor-23 (FGF-23) is a biomarker for the monitoring of chronic kidney disease in humans. The clinical relevance of FGF-23 in dogs is largely unknown. The aim of this study was (1) to show the intra- and interassay precision of the Kainos ELISA FGF-23 kit, (2) to determine a reference interval for FGF-23 in dogs, and (3) to investigate the possible correlations of the FGF-23 concentrations with other hematological and biochemical parameters. The coefficient of variation was <15% for both the intra- and interassay precision. The reference interval ranged between 95.8 (90% confidence interval: 44.6; 139.2) and 695.1 pg/mL (598.7; 799.1) in 136 clinically healthy dogs. For the correlation analysis, four groups were retrospectively formed based on the creatinine concentration classification according to the IRIS guidelines, including 10 dogs each. Strong positive correlations were detected between the FGF-23 concentration and the renal parameters. Statistically significant differences in the FGF-23 concentrations were demonstrated between study groups I and III (*p <* 0.001), I and IV (*p <* 0.001), and II and IV (*p* = 0.005).

**Abstract:**

Fibroblast growth factor-23 (FGF-23) is a phosphaturic hormone used to monitor chronic kidney disease (CKD) in humans. The aims of this study were (1) to determine the intra- and interassay precision of the FGF-23 concentrations in dogs as measured via the Kainos ELISA FGF-23 kit, (2) to calculate a reference interval, and (3) to assess the correlation of the FGF-23 concentration with the hematological and biochemical parameters. The coefficient of variation was below 15% for both the intra- and interassay precision, indicating good reproducibility. The reference interval ranged between 95.8 (90% confidence interval: 44.6; 139.2) and 695.1 pg/mL (598.7; 799.1) based on 136 clinically healthy dogs, classified as such according to the information of treating veterinarians as well as the unremarkable results of hematology and biochemistry. The FGF-23 concentration differed significantly between dogs aged <9 and ≥9 years (*p* = 0.045). Four groups of 10 dogs each were retrospectively formed based on the creatinine concentration classification according to the IRIS staging. Correlation was the strongest for the renal parameters. Statistically significant differences in the FGF-23 concentration were demonstrated between the study groups I and III (*p <* 0.001), I and IV (*p <* 0.001), and II and IV (*p* = 0.005). There was a trend for a rising FGF-23 concentration in older dogs. Due to the wide reference interval, diagnostic cut-offs and/or subject-based FGF-23 reference values in each dog are needed for monitoring and clinical interpretation.

## 1. Introduction

Chronic kidney disease (CKD) is a common, irreversible, and progressive disease in dogs. In the United States of America, CKD is reported to have a prevalence of up to 25% among the dogs presented to veterinary referrals [1,2]. Due to decreased urinary excretion, hyperphosphatemia is common in CKD leading to the renal secondary hyperparathyroidism [3,4,5] seen in approximately 76% of dogs with CKD [3]. The prognosis is based on staging according to the International Renal Interest Society (IRIS) guidelines, which is based on the serum creatinine concentration, blood pressure measurement, and urine protein/creatinine ratio [6]. The modified IRIS guidelines also include the serum symmetric dimethylarginine (SDMA) concentration [6].

Renal biomarkers available in dogs include endogenous markers of the glomerular filtration rate such as serum creatinine, symmetric dimethylarginine (SDMA), cystatin C, homocysteine, and neutrophil gelatinase-associated lipocalin, as well as the markers of altered metabolism, e.g., fibroblast growth factor-23 (FGF-23) [7].

FGF-23 is a phosphatonin produced by osteocytes and osteoblasts and plays a role in phosphorus and calcitriol homeostasis [8,9]. FGF-23 promotes the excretion of phosphorus via urine by reducing the expression of the sodium phosphate cotransporter in the kidneys and inhibiting the calcitriol synthesis. This results in a decreased phosphate concentration in the peripheral blood [10]. An increasing FGF-23 concentration was observed in humans and cats with progressing CKD [11,12,13]. In humans with CKD, an increase in the FGF-23 concentration was noted prior to any other changes in the other markers of mineral metabolism [14,15]. Additionally, an increase in the FGF-23 concentration was associated with poor prognosis in both humans and cats with CKD [16,17,18]. In dogs, FGF-23 was associated with the duration of survival in CKD [19].

Previous studies reported an association between FGF-23 and canine CKD; however, the aforementioned studies were based on a rather small number of dogs (34 and 42 dogs, respectively) [20,21]. Another study of 90 dogs, including a control group of healthy animals (*n* = 15), as well as dogs classified in stages 1 to 4 according to the IRIS guidelines (*n* = 75), suggested a positive correlation between the FGF-23 concentration and the IRIS stages [22]. Similar results were also reported in humans [13,14,23] and in cats in a study based on 40 animals [24]. In dogs with azotemic CKD, an increased FGF-23 concentration was associated with an increased risk of premature death alongside other previously identified prognostic markers such as proteinuria, hyperphosphatemia, advanced CKD stage, and body condition score [19].

No reference intervals for the FGF-23 concentrations and only limited data for the clinical value of FGF-23 in dogs are currently available. The aims of our study were (1) to calculate the intra- and interassay precision for intact FGF-23 as measured via the Kainos ELISA FGF-23 kit, (2) to calculate a reference interval for FGF-23 based on 140 healthy dogs, and (3) to assess the FGF-23 correlation with the hematological and biochemical parameters in 40 dogs classified based on their creatinine levels following the IRIS guidelines.

## 2. Materials and Methods

In the retrospective study, the reference intervals for FGF-23 were calculated according to the ASVCP reference interval guidelines [25] using the reference interval advisor software 2.1 on Microsoft Excel for Microsoft Windows 365 [26]. The samples for the calculation of the reference intervals were collected as left-over samples from send-in routine diagnostic cases to the clinical laboratory LABOKLIN (Bad Kissingen, Germany). The samples qualified for the study had to include a Complete Blood Count (CBC) from EDTA blood using the Sysmex XN-V analyzer (Sysmex Deutschland, Norderstedt, Germany) and a biochemistry profile measured from the serum samples on Cobas 8000 (Roche, Mannheim, Germany), including alanine transaminase, alkaline phosphatase, bilirubin, triglycerides, cholesterol, alpha-amylase, DGGR-lipase, glucose, fructosamine, urea, creatinine, symmetric dimethylarginine, magnesium, phosphorus, potassium, sodium, calcium, total protein, albumin, globulin, glutamate dehydrogenase, and creatine kinase. Only dogs with unremarkable hematological and biochemical results according to the reference intervals of the laboratory were included. The general guidelines of the laboratory recommend taking blood samples in dogs after being fasted overnight. Hematological and biochemical parameters were evaluated immediately after an overnight shipment of the serum samples to the laboratory. For the FGF-23 analysis, the samples were stored frozen for a maximum of 5 working days at −20 °C. The submitting veterinarians of suitable sent-in clinical cases were asked to provide information about the general health status of the individual dogs at the time of blood collection, whether any concurrent chronic diseases were known, whether there was suspicion of CKD, and whether the dogs were fed a renal diet. The data were collected via questionnaires and telephone calls to the veterinarians submitting the samples. Only dogs were included in the study for the determination of reference intervals that were generally assessed as clinically healthy by the treating veterinarian, did not suffer from any known acute or chronic diseases, were not suspicious to have or being developing a CKD, and did not receive a specific renal diet. The FGF-23 concentration was measured in serum with the FGF-23 ELISA Kit (Kainos Laboratories, Tokyo, Japan) in all the dogs that were classified in the described way as “healthy”. Originally, this assay was used to detect human FGF-23 but was already successfully used for the detection in dogs [20]. According to the manufacturer’s specifications, the minimum detection limit for the Kainos ELISA assay was 3 pg/mL, while the quantification range spanned 3–800 pg/mL. The highest concentration of the standard curve of this ELISA is 800 pg/mL. In order to cover a wider dynamic range of results, the samples were diluted 1:2 with a diluent provided by the ELISA manufacturer and the dilution was taken into account for the calculation of the respective results of the measurements. This means that for the measurement of FGF-23, the highest point of the standard curve practically lay at 1600 pg/mL, where the values above this point were extrapolated from the standard curve; due to this, the values above 4000 pg/mL were capped to >4000 pg/mL [27]. Prior transformation of the reference data was performed and Horn’s algorithm using Turkey’s interquartile fences were used for the identification of potential outliers.

For both the intraassay and interassay precision, the coefficient of variation (CV) was calculated. For the intraassay precision, five FGF-23 concentration measurements in the samples of three different concentrations (low, moderate, and high) were performed within a single run. For the interassay precision, the FGF-23 concentration was measured in the samples with three different FGF-23 concentrations (low, moderate, and high) of the same sample once a day for five consecutive days. The samples were stored frozen at −20 °C for a maximum of five days.

For the correlation analysis of the hematological and biochemical parameters, another study population was retrospectively selected using the leftover samples from routine diagnostics. Four study groups were formed according to the creatinine concentration classification via the IRIS guidelines (study group I–IV: creatinine concentration according to IRIS 1–4, *n* = 10 dogs each) [6]. The health status was not determined in these 40 dogs. The FGF-23 concentration was measured using the FGF-23 ELISA Kit. Hematological examination was performed using the Sysmex XN-V analyzer (Sysmex Deutschland, Norderstedt, Germany), while the biochemistry profile was measured on Cobas 8000 (Roche, Mannheim, Germany) (Table 1, Appendix A). Spearman’s rank correlation coefficient was used to calculate the correlation between the FGF-23 concentration and the hematological, as well as few renal parameters, if available.

Descriptive statistical analysis was performed using SPSS for Windows (version 28.0; International Business Machines Corporation, Armonk, NY, USA. *p* < 0.05 was considered statistically significant. The Shapiro–Wilk test was used for the assessment of normal distribution. The Kruskal–Wallis test was used to calculate the statistical significance between the study groups. Bonferroni correction was applied where indicated. Spearman’s rank correlation coefficient (*p*) was also calculated. *p* < 0.2 was classified as a very weak correlation, *p* = 0.2–0.5 as a weak correlation, *p* > 0.5–0.7 as a moderate correlation, *p* > 0.7–0.9 as a strong correlation, and *p* > 0.9 as a very strong correlation.

The study was conducted using the leftover material from the routine samples submitted to the commercial laboratory LABOKLIN (Bad Kissingen, Germany) by veterinarians in Germany. According to the terms and conditions of the laboratory LABOKLIN as well as the RUF-55.2.2.2532-1-86-5 decision of the government of Lower Franconia, no special permission from the animal owners or the animal welfare commission is needed for the additional testing on the residual sample material once the diagnostics are completed.

## 3. Results

The intraassay precision of the Kainos ELISA assay for the FGF-23 measurements in dogs showed a CV of 5.62%, 8.59%, and 5.13%, while the interassay precision was 9.96%, 7.71%, and 10.82% for the samples with low, moderate, and high FGF-23 levels, respectively.

One hundred and forty clinically healthy dogs based on the unremarkable results in the CBC and biochemistry testing as well as based as on the results of the questionnaires were initially considered for the reference interval calculation of FGF-23, out of which four (2.9%) were identified as outliers (Figure 1 and Figure 2). Of the remaining 136 dogs, 58 were males (42.6%, 23/58 intact [39.7%], 35/58 castrated [60.3%]) and 78 females (57.5%, 49/78 intact [62.8%], 29/78 spayed [37.2%]). The breed was known in 130/136 dogs (95.6%), including 45 different breeds (most often mixed breed (*n* = 51, 39.2%), Labrador–Retriever (*n* = 12, 9.2%), and Chihuahua (*n* = 4, 3.1%)). The age was known in all 136 dogs (mean 8.2 years, standard deviation 3.4 years, median 9.0 years, minimum 1 year, maximum 16 years).

The results for the FGF-23 concentration were available for all 136 dogs (mean 304.2 pg/mL, median 297.8 pg/mL, standard deviation 118.6 pg/mL, minimum 44.63 pg/mL, and maximum 799.1 pg/mL), and the data were normally distributed (*p* = 0.054). In the non-parametric analysis, the lower limit of the reference interval was 95.8 pg/mL [90% CI: 44.6; 139.2] and the upper limit 695.1 pg/mL [90% CI: 598.7; 799.1] (Figure 1). In dogs of <9 years of age (68/140, 48.6%), the reference interval was calculated using robust methods and ranged from 83.5 pg/mL [66.4; 105.7] to 709.6 pg/mL [596.3; 839.0], while in dogs of ≥9 years of age (72/140, 51.4%), it ranged from 128.8 pg/mL [108.6; 154.8] to 635.0 pg/mL [572.5; 693.7]. Statistically significant differences in the FGF-23 concentrations between both age categories were demonstrated (χ^2^ = 3.897, *df* = 1, *p* = 0.048) (Figure 3).

In the whiskers diagram, 50% of the reference individuals were distributed between 224.0 and 404.3 pg/mL regarding the FGF-23 concentrations (Figure 2).

In the second part of the study, we analyzed potential correlations between the FGF-23 concentrations and other hematological markers as well as the markers of clinical chemistry. A total of forty dogs were included in which the health status was not known in the form of direct information from the treating veterinarian. The following signalment information for each dog was collected: age (available for all 40 dogs or 100.0%, mean 7.1 years, standard deviation 4.8 years, median 7.0 years, minimum 1 year, maximum 17 years), breed (available for 37/40 dogs or 92.5%; mixed breeds (*n* = 11, 27.5%), Dachshund (*n* = 5, 12.5%), Australian Shepard (*n* = 3, 7.5%), Labrador–Retriever and Yorkshire Terrier (*n* = 2 each, 5.0%), Appenzell Mountain Dog, American Bulldog, Bobtail, Bolonka Zvetna, Boxer, Epagneul Breton, Chihuahua, German spitz dog, Elo, Havanese, Labradoodle, Maltese, Swiss mountain dog, and Springer Spaniel (*n* = 1 each, 2.5%), and sex (available for all 40 dogs or 100.0%; 23/40 dogs or 57.5% were male, intact *n* = 15, castrated *n* = 8; 17/40 dogs or 42.5% were female, non-spayed *n* = 11, spayed *n* = 6).

The results for the FGF-23 measurements were available in all 40 dogs (Figure 4), for the hematological parameters in 36 dogs (90.0%), while for the serum biochemistry, the available parameters varied due to the insufficient sample material for performing all analyses (Appendix A). The FGF-23 concentration was normally distributed in study groups I (*p* = 0.391), II (*p* = 0.480), and III (*p* = 0.593), while a non-normal distribution was noted in study group IV (*p* = 0.001) (Appendix A). The FGF-23 concentrations differed significantly between the study groups (χ^2^ = 32.539, *df* = 3, *p* < 0.001) (Table 1). A statistically significant difference in the FGF-23 concentration measurements was demonstrated between study groups I and III (*p* < 0.001), I and IV (*p* < 0.001), and II and IV (*p* = 0.005). Meanwhile, no statistically significant differences were seen between study groups I and II (*p* = 0.397), II and III (*p* = 0.120), and III and IV (*p* = 1.000).

In hematology, when comparing the individual study groups, statistically significant differences were detected for the red blood cell count, hematocrit, and absolute numbers of basophils (Table 1). Spearman’s rank correlation coefficient analysis revealed a low negative correlation (*p* = −0.363, *p* = 0.030) between the FGF-23 concentration and both relative and absolute amounts of basophils, as well as for reticulocyte hemoglobin (*p* = −0.373, *p* = 0.025). No statistically significant difference in the FGF-23 concentration was observed for the red blood cell count (*p* = -0.316, *p* = 0.061) and hematocrit (*p* = −0.320, *p* = 0.057) across the individual study groups.

Regarding the biochemical parameters (Table 2), a strong positive correlation with the FGF-23 concentration was detected for creatinine (*p* = 0.899), urea (*p* = 0.831) and phosphorus (*p* = 0.734) (*p* < 0.001, each). A moderate positive correlation was seen for alpha amylase (*p* = 0.548), DGGR-lipase (*p* = 0.518) and magnesium (*p* = 0.521) (*p* < 0.001, each). A weak positive correlation was demonstrated for bilirubin (*p* = 0.458, *p* = 0.003) and globulin (*p* = 0.418, *p* = 0.007) (Table 3).

## 4. Discussion

This is the first study providing reference intervals for the FGF-23 concentration measurements with Kainos ELISA in dogs. A wide reference interval ranging from 95.8 pg mL [44.6 pg/mL; 139.2 pg/mL] to 695.1 pg/mL [598.7 pg/mL; 799.1 pg/mL] was calculated. The 90% Cis of the lower and upper limit were broader than recommended in IFCC-CLSI C28-A3, emphasizing the need for diagnostic cut-offs when assessing the FGF-23 concentration in dogs. Additionally, the results also indicated the need for subject-based FGF-23 reference values in healthy dogs in order to detect and monitor the pathophysiological changes in the FGF-23 concentration. The calculated intra- and interassay CVs for the precision assessment for Kainos ELISA were all <15%, which was consistent with the previous studies assessing the FGF-23 measurements with the Kainos ELISA assay when measuring intact FGF-23 [20] and indicated a good test performance. The storage time of a maximum of five days of frozen samples In our study most likely did not influence the FGF-23 concentrations, as FGF-23 was stable in the samples stored at different temperatures and lengths of time without significant changes in the earlier studies [29,30].

In dogs included in the calculation of the reference intervals, the health status was based both on the results of the questionnaires to the veterinarians as well as on the unremarkable hematological and biochemical results, which represent strong inclusion and exclusion criteria. Potentially important background information was unavailable to the authors, including the living conditions and reasons for the blood sampling and laboratory testing. Therefore, no data regarding history, clinical signs, urinalysis, diagnostic imaging, and prior or current medications were included in the study and the exclusion of pre- or post-renal azotemia was not explicitly possible. Especially, the application of phosphate binders could not be ruled out with certainty.

There has been an indication that the FGF-23 concentration might increase with age in clinically healthy cats [28]. This trend was also seen in the dogs classified as clinically healthy based on the questionnaires, where a statistically significant difference in the FGF-23 concentrations was noted when comparing dogs aged <9 years to dogs of ≥9 years of age (*p* = 0.048). Possible reasons for this finding are currently unclear but must be kept in mind when interpreting the FGF-23 results.

In the second part of our study, including 40 dogs for the correlation analysis, a strong positive correlation was noted for the renal parameters, namely creatinine, urea, and phosphorus (*p* > 0.700, *p* < 0.001 each). A similar correlation with creatinine and phosphorus was also observed in another study using the same Kainos ELISA assay [20]. Statistically significant differences in the FGF-23 concentrations were demonstrated between study groups I and III (*p* < 0.001), I and IV (*p* < 0.001), and II and IV (*p* = 0.005). In general, the identification of an IRIS stage I group (renal disease present but not yet azotemic) is impossible without further diagnostic information. Study group I therefore is not synonymous to IRIS 1 and dogs might not have had early kidney disease and instead might have represented healthy dogs. Additionally, it must be considered that CKD was not confirmed in the groups II to IV in our study, as the classification was exclusively based on the creatinine concentration. The inclusion of azotemic dogs without any background knowledge regarding the underlying causes represents an important limitation. Although the laboratory recommends taking blood samples in dogs fasted overnight, information regarding the fasting status of the dogs included in the study could not be obtained due to the retrospective study design. Therefore, a potential influence of the postprandial state and diet composition on the concentrations of the parameters of renal dysfunction and the FGF-23 concentrations must be considered. In humans, e.g., postprandial influence on the FGF-23 concentrations was demonstrated after 12 h from a meal [31]. The correlations of the FGF-23 concentrations with the renal parameters and phosphorus in this study therefore need to be judged with caution.

Meanwhile, no statistically significant differences were seen between study groups I and II (*p* = 0.397), II and III (*p* = 0.120), and III and IV (*p* = 1.000). Similar results were also noted in a study of the FGF-23 concentration measurements in 32 dogs, noting a statistically significant difference between 16 dogs of IRIS stage I and II and the other 16 dogs of IRIS stage III and IV (*p* < 0.001) [20]. An additional limitation was the low number of animals in each study group. An IRIS staging according to the guidelines with serum creatinine and the SDMA concentration measurements, as well as with UPC and a blood pressure assessment, is further recommended [6].

Overall, the significance of FGF-23 as a biomarker remains unclear, while the preliminary findings indicate that this parameter could be potentially used for the CKD monitoring, taking the creatinine concentrations in consideration. FGF-23 seems less suitable for diagnosing CKD though. It is also suggested that FGF-23 may be suboptimal for the early detection of CKD or the monitoring of early-stage kidney disease. Creatinine is an indirect biomarker of the glomerular filtration rate. The FGF-23 concentration is known to be inversely related to GFR in humans [32]. Therefore, it was not surprising that the highest correlation of all biochemical parameters was detected between the FGF-23 and creatinine concentrations in our study. A similar finding was also noted in other studies [20].

Our findings are in concordance with another study [20], demonstrating only a moderate correlation between the FGF-23 concentrations and serum phosphorus. Due to the classification of FGF-23 as a phosphatonin, a stronger correlation was expected, which may be explained using the higher FGF-23 concentrations resulting in the phosphorous concentrations being kept within the reference interval in the early stage of the disease. This was also noted in cats using the Kainos ELISA assay when only a moderate correlation between FGF-23 and phosphorus was observed [24]. In studies using the MyBioSource ELISA assay in cats (MyBioSource, San Diego, CA, USA), no statistically significant correlation between the FGF-23 concentration and serum phosphorus was seen [24,33]. The phosphate levels may be comparatively low in patients with azotemia; therefore, this finding should be carefully investigated in further studies where the clinical background information is more available. In general, the correlation findings between FGF-23 and phosphorus may indicate that the correlation depends on the diagnostic assay chosen for the FGF-23 concentration measurements. Furthermore, FGF-23 may increase prior to hyperphosphatemia to avoid increases in the phosphorus concentrations.

A moderate, statistically significant positive correlation was demonstrated between the magnesium and FGF-23 concentrations in 37 dogs. In humans with CKD, disturbances of magnesium homeostasis are rather common and are associated with increased mortality. A significant correlation was detected between FGF-23, the fractional excretion of magnesium, and urinary magnesium, but not with the plasma magnesium concentrations [34]. In cats with azotemic CKD, hypomagnesemia was associated with a higher plasma FGF-23 concentration and an increased risk of mortality [35]. The reason for the positive correlation of serum magnesium levels with the FGF-23 concentrations in the dogs in our study remains unclear. Prospective studies with a larger study population are recommended to better understand the correlation between FGF-23 and magnesium.

Chronic inflammatory disease can contribute to the activation of osteocytes by circulating cytokines and leads to the increased plasma concentration of FGF-23. Proinflammatory cytokines such as TNFα, IL-1β, and IL-6 are potent stimuli for the FGF-23 secretion by osteocytes and have been demonstrated to play a role in upregulating intact FGF23 in experimental models in vivo of both acute kidney injury and chronic kidney disease [36]. A moderate, positive, statistically significant correlation was demonstrated between both DGGR-lipase as well as the alpha amylase concentrations and FGF-23 levels. As increased DGGR lipase activity in dogs is indicative for pancreatitis, further studies for the relationship between CKD-induced FGF-23 elevation and inflammatory diseases such as pancreatitis in dogs are needed.

Iron deficiency is an important factor promoting anemia in humans with CKD. A study with mice demonstrated an increase in FGF-23 production when absolute and functional iron deficiency was present [37]. In dogs, low reticulocyte hemoglobin levels are indicative for iron deficiency, even if all other hematological parameters were within the reference intervals [38]. Our data show a low negative correlation between the FGF-23 concentration and reticulocyte hemoglobin, supporting the findings in humans and in mice to be possibly relevant in dogs as well. However, prospective studies with larger study cohorts in dogs are needed for a further clarification of the correlation between the reticulocyte hemoglobin levels and FGF-23 concentrations.

## 5. Conclusions

The intra- and interassay precision for the FGF-23 measurements in dogs using the Kainos ELISA assay demonstrated CVs of <15%, indicating good reproducibility. The FGF-23 concentration can differentiate between the early and late IRIS CKD stages, but the relevance of FGF-23 as an early biomarker for detecting CKD remains unclear. The wide reference interval for FGF-23 is indicative that subject-based FGF-23 reference values should be considered in healthy dogs in order to detect and monitor the pathophysiological changes in the FGF-23 levels and/or CKD. Diagnostic cut-offs are needed for the clinical interpretation. FGF-23 may be suitable for the CKD monitoring. However, the diagnostic relevance of FGF-23 in dogs needs to be investigated further. There was a trend for a rising FGF-23 concentration in older dogs.

## Figures and Tables

**Figure 1 animals-13-03202-f001:**
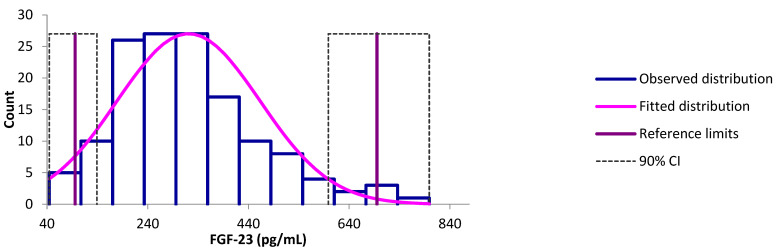
Schematic representation of a reference interval, reference limits, and 90% confidence intervals (CI) of the limits for FGF-23 concentration as measured via the FGF-23 ELISA Kit (Kainos Laboratories, Tokyo, Japan) in 140 clinically healthy dogs classified as such according to questionnaires to the veterinarians and unremarkable hematological as well as biochemical results.

**Figure 2 animals-13-03202-f002:**
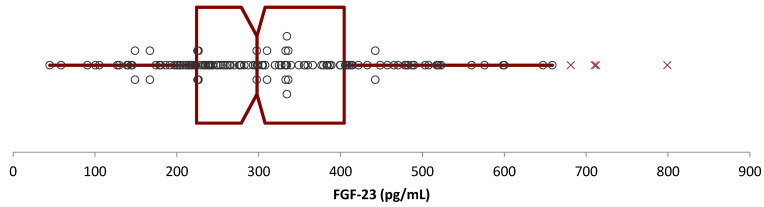
Printout of computations made with a series of 140 canine serum FGF-23 concentrations using Reference Value Advisor. The four outliers detected via Tukey’s test appear as red crosses on the whiskers diagram.

**Figure 3 animals-13-03202-f003:**
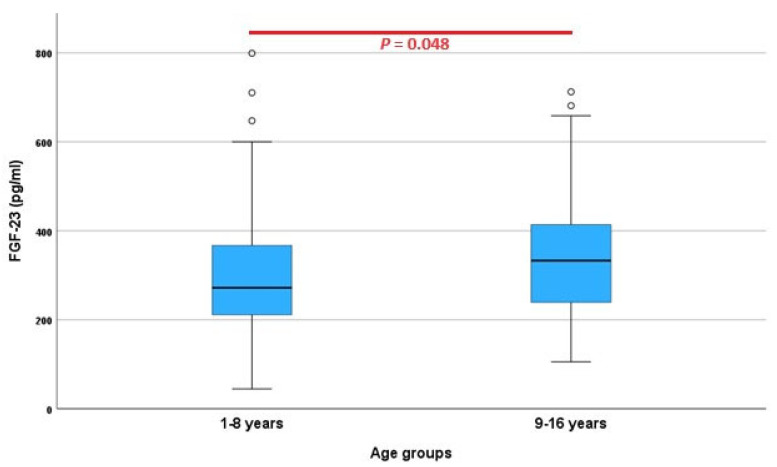
Serum fibroblast growth factor (FGF)-23 concentration as measured via the FGF-23 ELISA Kit (Kainos Laboratories, Tokyo, Japan) in 140 clinically healthy dogs divided in two age categories (1–8 years: *n* = 68; 48.6%; 9–16 years: *n* = 72, 51.4%). ° = mild outliers (values that are more than 1.5 × interquartile range below Q1 or above Q3 in the boxplot).

**Figure 4 animals-13-03202-f004:**
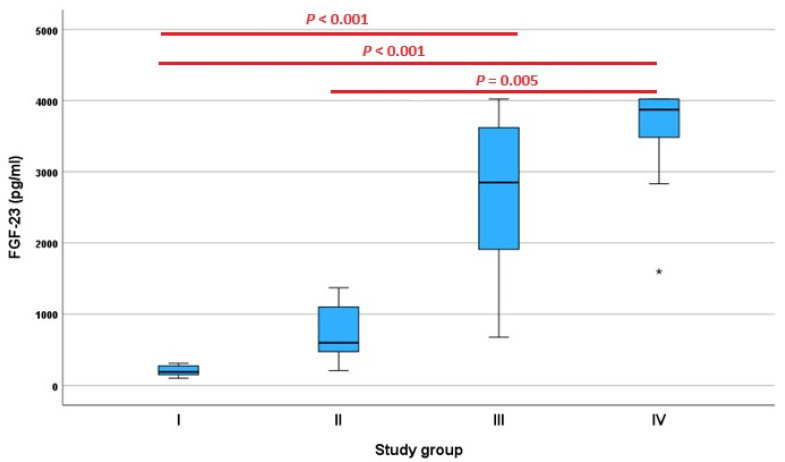
Fibroblast growth factor (FGF)-23 concentration as measured via the FGF-23 ELISA Kit (Kainos Laboratories, Tokyo, Japan) in 40 dogs divided into study groups I, II, III, and IV (*n* = 10 each) based on creatinine concentration according to the guidelines of the International Renal Interest Society (IRIS). * = extreme outliers (values that are more than 3.0 × interquartile range below Q1 or above Q3 in the boxplot).

**Table 1 animals-13-03202-t001:** Age, results of a Complete Blood Count and fibroblast growth factor (FGF)-23 concentrations in dogs when divided in groups I–IV based on creatinine concentrations according to the guidelines of the International Renal Interest Society (IRIS) (*N* tested dogs, mean (M), median (ME), standard deviation (SD), minimum (Min), maximum (Max)).

Study Group	RI	I	II	III	IV	*p* ^1^
Age (years)	-	*N* = 10,	*N* = 10,	*N* = 10,	*N* = 10,	-
		M = 2.1,	M = 10.5,	M = 8.8,	M = 6.8,	
		ME = 1.5,	ME = 11.0,	ME = 8.0,	ME = 6.0,	
		SD = 1.4,	SD = 3.2,	SD = 4.2,	SD = 5.0,	
		Min = 1.0,	Min = 5.0,	Min = 3.0,	Min = 1.0,	
		Max = 5.0	Max = 14.0	Max = 17.0	Max = 17.0	
FGF-23 (pg/mL) ^A^	56–700 ^2^	*N* = 10,	*N* = 10,	*N* = 10,	*N* = 10,	χ^2^ = 32.539, *df* = 3, *p* < 0.001
		M = 203.0,	M = 735.5,	M = 2688.5,	M = 3532.5,	
		ME = 189.2,	ME = 597.7,	ME = 2848.9,	ME = 3873.1,	
		SD = 74.6,	SD = 397.7,	SD = 574.1,	SD = 778.5,	
		Min = 102.0,	Min = 208.6,	Min = 675.1,	Min = 1597.7,	
		Max = 309.1	Max = 1370.1	Max = 4022.2	Max = 4022.2	
Hematology ^B^						
Red blood cell count (×10^9^/L)	5.5–8.5 ^3^	*N* = 10,	*N* = 9,	*N* = 7,	*N* = 10,	χ^2^ = 9.578, *df* = 3, *p* = 0.023
		M = 6.2,	M = 7.1,	M = 5.8,	M = 4.7,	
		ME = 6.0,	ME = 7.0,	ME = 5.1,	ME = 4.7,	
		SD = 1.8,	SD = 0.9,	SD = 2.0,	SD = 1.4,	
		Min = 2.1,	Min = 5.9,	Min = 3.1,	Min = 3.2,	
		Max = 8.5	Max = 8.8	Max = 8.9	Max = 7.5	
Hemoglobin (g/L)	150.0–190.0 ^3^	*N* = 10,	*N* = 9,	*N* = 7,	*N* = 10,	χ^2^ = 6.764, *df* = 3, *p* = 0.080
		M = 148.6,	M = 165.3,	M = 135.1,	M = 119.1,	
		ME = 145.5,	ME = 170.0,	ME = 130.0,	ME = 116.0,	
		SD = 42.9,	SD = 22.8,	SD = 42.8,	SD = 37.0,	
		Min = 49.0,	Min = 132.0,	Min = 76.0,	Min = 77.0,	
		Max = 201.0	Max = 193.0	Max = 193.0	Max = 192.0	
Hematocrit (L/L)	0.44–0.52 ^3^	*N* = 10,	*N* = 9,	*N* = 7,	*N* = 10,	χ^2^ = 9.660, *df* = 3, *p* = 0.022
		M = 0.43,	M = 0.48,	M = 0.38,	M = 0.34,	
		ME = 0.45,	ME = 0.49,	ME = 0.38,	ME = 0.34,	
		SD = 0.12,	SD = 0.06,	SD = 0.11,	SD = 0.09,	
		Min = 0.14,	Min = 0.40,	Min = 0.22,	Min = 0.23,	
		Max = 0.54	Max = 0.57	Max = 0.54	Max = 0.51	
Reticulocytes (/µL)	<110.0 ^3^	*N* = 10,	*N* = 9,	*N* = 7,	*N* = 10,	χ^2^ = 6.536, *df* = 3, *p* = 0.088
		M = 44.6,	M = 80.0,	M = 63.8,	M = 69.2,	
		ME = 31.6,	ME = 81.5,	ME = 63.3,	ME = 52.7,	
		SD = 26.7,	SD = 26.9,	SD = 31.1,	SD = 51.8,	
		Min = 14.9,	Min = 46.4,	Min = 30.6,	Min = 20.5,	
		Max = 95.6	Max = 134.0	Max = 108.9	Max = 165.8	
Reticulocyte hemoglobin (pg)	>20.1 ^3^	*N* = 10,	*N* = 9,	*N* = 7,	*N* = 10,	χ^2^ = 7.636, *df* = 3, *p* = 0.054
		M = 25.3,	M = 24.9,	M = 24.7,	M = 27.1,	
		ME = 25.9,	ME = 25.5,	ME = 26.5,	ME = 27.3,	
		SD = 1.7,	SD = 2.0,	SD = 4.0,	SD = 1.5,	
		Min = 22.4,	Min = 20.5,	Min = 18.0,	Min = 24.6,	
		Max = 27.0	Max = 27.4	Max = 28.7	Max = 29.2	
White blood cell count (×10^12^/L)	6.0–12.0 ^3^	*N* = 10,	*N* = 9,	*N* = 7,	*N* = 10,	χ^2^ = 4.270, *df* = 3, *p* = 0.234
		M = 11.3,	M = 8.4,	M = 14.0,	M = 11.2,	
		ME = 11.3,	ME = 8.1,	ME = 15.8,	ME = 9.7,	
		SD = 4.1,	SD = 2.6,	SD = 7.1,	SD = 5.1,	
		Min = 6.0,	Min = 5.5,	Min = 6.2,	Min = 7.3,	
		Max = 18.2	Max = 13.7	Max = 25.6	Max = 24.4	
Neutrophils (%)	55.0–75.0 ^3^	*N* = 10,	*N* = 9,	*N* = 7,	*N* = 10,	χ^2^ = 2.035, *df* = 3, *p* = 0.565
		M = 68.7,	M = 69.3,	M = 71.7,	M = 77.3,	
		ME = 68.5,	ME = 66.0,	ME = 68.0,	ME = 80.0,	
		SD = 16.4,	SD = 9.0,	SD = 16.8,	SD = 11.0,	
		Min = 46.0,	Min = 56.0,	Min = 50.0,	Min = 59.0,	
		Max = 87.0	Max = 84.0	Max = 93.0	Max = 94.0	
Neutrophils (×10^12^/L)	3.0–9.0 ^3^	*N* = 10,	*N* = 9,	*N* = 7,	*N* = 10,	χ^2^ = 3.518, *df* = 3, *p* = 0.318
		M = 7.9,	M = 5.9,	M = 10.7,	M = 8.9,	
		ME = 7.2,	ME = 5.2,	ME = 8.4,	ME = 7.4,	
		SD = 3.5,	SD = 2.2,	SD = 7.0,	SD = 5.1,	
		Min = 3.0,	Min = 3.6,	Min = 3.1,	Min = 4.8,	
		Max = 14.1	Max = 9.5	Max = 22.0	Max = 22.0	
Eosinophils (%)	<6.0 ^3^	*N* = 10,	*N* = 9,	*N* = 7,	*N* = 10,	χ^2^ = 3.072, *df* = 3, *p* = 0.381
		M = 2.8,	M = 5.1,	M = 3.9,	M = 3.7,	
		ME = 1.0,	ME = 4.0,	ME = 4.0,	ME = 2.0,	
		SD = 3.4,	SD = 4.1,	SD = 2.9,	SD = 3.3,	
		Min = 0.0,	Min = 1.0,	Min = 0.0,	Min = 1.0,	
		Max = 11.0	Max = 13.0	Max = 8.0	Max = 10.0	
Eosinophils (×10^12^/L)	0.04–0.6 ^3^	*N* = 10,	*N* = 9,	*N* = 7,	*N* = 10,	χ^2^ = 1.908, *df* = 3, *p* = 0.592
		M = 0.3,	M = 0.4,	M = 0.4,	M = 0.4,	
		ME = 0.1,	ME = 0.3,	ME = 0.4,	ME = 0.3,	
		SD = 0.4,	SD = 0.3,	SD = 0.2,	SD = 0.3,	
		Min = 0.0,	Min = 0.1,	Min = 0.0,	Min = 0.1,	
		Max = 1.1	Max = 1.1	Max = 0.6	Max = 1.1	
Lymphocytes (%)	13.0–30.0 ^3^	*N* = 10,	*N* = 9,	*N* = 7,	*N* = 10,	χ^2^ = 1.866, *df* = 3, *p* = 0.601
		M = 21.5,	M = 19.8,	M = 18.6,	M = 14.7,	
		ME = 18.5,	ME = 22.0,	ME = 20.0,	ME = 13.5,	
		SD = 14.9,	SD = 6.7,	SD = 12.8,	SD = 8.7,	
		Min = 3.0,	Min = 7.0,	Min = 3.0,	Min = 4.0,	
		Max = 44.0	Max = 26.0	Max = 36.0	Max = 31.0	
Lymphocytes (×10^12^/L)	1.0–3.6 ^3^	*N* = 10,	*N* = 9,	*N* = 7,	*N* = 10,	χ^2^ = 1.669, *df* = 3, *p* = 0.644
		M = 2.4,	M = 1.6,	M = 2.1,	M = 1.5,	
		ME = 1.9,	ME = 1.4,	ME = 1.8,	ME = 1.3,	
		SD = 1.7,	SD = 0.7,	SD = 1.7,	SD = 0.7,	
		Min = 0.2,	Min = 0.5,	Min = 0.5,	Min = 0.3,	
		Max = 6.0	Max = 3.0	Max = 5.7	Max = 2.5	
Monocytes (%)	0.0–4.0 ^3^	*N* = 10,	*N* = 9,	*N* = 7,	*N* = 10,	χ^2^ = 4.681, *df* = 3, *p* = 0.197
		M = 6.7,	M = 5.8,	M = 5.9,	M = 4.3,	
		ME = 7.0,	ME = 5.0,	ME = 5.0,	ME = 4.0,	
		SD = 2.3,	SD = 2.0,	SD = 3.2,	SD = 2.8,	
		Min = 2.0,	Min = 4.0,	Min = 2.0,	Min = 1.0,	
		Max = 10.0	Max = 9.0	Max = 12.0	Max = 9.0	
Monocytes (×10^12^/L)	0.04–0.5 ^3^	*N* = 10,	*N* = 9,	*N* = 7,	*N* = 10,	χ^2^ = 4.716, *df* = 3, *p* = 0.194
		M = 0.78,	M = 0.48,	M = 0.76,	M = 0.46,	
		ME = 0.65,	ME = 0.50,	ME = 0.70,	ME = 0.40,	
		SD = 0.45,	SD = 0.16,	SD = 0.47,	SD = 0.33,	
		Min = 0.20,	Min = 0.20,	Min = 0.30,	Min = 0.10,	
		Max = 1.80	Max = 0.70	Max = 1.30	Max = 1.00	
Basophils (%)	0.0 ^3^	*N* = 10,	*N* = 9,	*N* = 7,	*N* = 10,	χ^2^ = 8.273, *df* = 3, *p* = 0.041
		M = 0.3,	M = 0.0,	M = 0.0,	M = 0.0,	
		ME = 0.0,	ME = 0.0,	ME = 0.0,	ME = 0.0,	
		SD = 0.5,	SD = 0.0,	SD = 0.0,	SD = 0.0,	
		Min = 0.0,	Min = 0.0,	Min = 0.0,	Min = 0.0,	
		Max = 1.0	Max = 0.0	Max = 0.0	Max = 0.0	
Basophils (×10^12^/L)	<0.04 ^3^	*N* = 10,	*N* = 9,	*N* = 7,	*N* = 10,	χ^2^ = 8.259, *df* = 3, *p* = 0.041
		M = 0.04,	M = 0.0,	M = 0.0,	M = 0.0,	
		ME = 0.0,	ME = 0.0,	ME = 0.0,	ME = 0.0,	
		SD = 0.07,	SD = 0.0,	SD = 0.0,	SD = 0.0,	
		Min = 0.0,	Min = 0.0,	Min = 0.0,	Min = 0.0,	
		Max = 0.2	Max = 0.0	Max = 0.0	Max = 0.0	
Thrombocytes (×10^12^/L)	150.0–500.0 ^3^	*N* = 10,	*N* = 9,	*N* = 7,	*N* = 10,	χ^2^ = 5.787, *df* = 3, *p* = 0.122
		M = 255.6,	M = 379.7,	M = 550.6,	M = 321.4,	
		ME = 249.5,	ME = 373.0,	ME = 495.0,	ME = 239.0,	
		SD = 86.3,	SD = 148.0,	SD = 310.8,	SD = 260.4,	
		Min = 130.0,	Min = 139.0,	Min = 130.0,	Min = 25.0,	
		Max = 428.0	Max = 611.0	Max = 913.0	Max = 929.0	

FGF-23 = Fibroblast growth factor-23; RI = reference interval. ^1^ Statistical differences between study groups I–IV, Kruskal–Wallis test with Bonferroni correction, *p* < 0.05 stated as statistically significant; ^2^ reference range is based on [28]); ^3^ reference ranges were based on the internal values of the LABOKLIN laboratory (Bad Kissingen, Germany); ^A^ FGF-23 ELISA Kit, Kainos Laboratories, Tokyo, Japan; ^B^ Sysmex XT 2000iv, Sysmex Deutschland, Germany.

**Table 2 animals-13-03202-t002:** Biochemical parameters and fibroblast growth factor (FGF)-23 concentrations in dogs when divided in groups I–IV based on creatinine concentrations according to the guidelines of the International Renal Interest Society (IRIS) (N tested dogs, mean (M), median (ME), standard deviation (SD), minimum (Min), maximum (Max)).

Study Group	RI	I	II	III	IV	*p* ^1^
FGF-23 (pg/mL) ^A^	56–700 ^2^	*N* = 10,	*N* = 10,	*N* = 10,	*N* = 10,	χ^2^ = 32.539, *df* = 3, *p* < 0.001
		M = 203.0,	M = 735.5,	M = 2688.5,	M = 3532.5,	
		ME = 189.2,	ME = 597.7,	ME = 2848.9,	ME = 3873.1,	
		SD = 74.6,	SD = 397.7,	SD = 574.1,	SD = 778.5,	
		Min = 102.0,	Min = 208.6,	Min = 675.1,	Min = 1597.7,	
		Max = 309.1	Max = 1370.1	Max = 4022.2	Max = 4022.2	
Biochemistry ^B^						
Creatinine (µmol/L)	35.0–106.0 ^3^	*N* = 10,	*N* = 10,	*N* = 10,	*N* = 10,	χ^2^ = 36.613, *df* = 3, *p* < 0.001
		M = 39.9,	M = 162.4,	M = 319.9,	M = 755.4,	
		ME = 39.5,	ME = 146.0,	ME = 308.0,	ME = 639.5,	
		SD = 4.0,	SD = 45.5,	SD = 53.4,	SD = 276.1,	
		Min = 31.0,	Min = 130.0,	Min = 265.0,	Min = 449.0,	
		Max = 45.0	Max = 248.0	Max = 430.0	Max = 1262.0	
Urea (mmol/L)	3.3–8.3 ^3^	*N* = 10,	*N* = 10,	*N* = 10,	*N* = 10,	χ^2^ = 31.595, *df* = 3, *p* < 0.001
		M = 5.0,	M = 14.9,	M = 38.7,	M = 66.6,	
		ME = 5.3,	ME = 11.3,	ME = 35.1,	ME = 61.3,	
		SD = 1.4,	SD = 12.1,	SD = 14.3,	SD = 20.6,	
		Min = 3.0,	Min = 5.1,	Min = 20.5,	Min = 39.3,	
		Max = 7.6	Max = 45.3	Max = 62.6	Max = 102.7	
Symmetric dimethylarginine (µmol/L)	<0.65	*N* = 0	*N* = 2,	*N* = 0	*N* = 1,	-
			M = 0.73,		M = 2.3,	
			ME = 0.73,		ME = 2.3,	
			SD = 0.04,		SD = 0.0	
			Min = 0.70,		Min = 2.3,	
			Max = 0.76		Max = 2.3	
Total calcium (mmol/L)	2.3–3.0 ^3^	*N* = 10,	*N* = 10,	*N* = 10,	*N* = 10,	χ^2^ = 3.038, *df* = 3, *p* = 0.386
		M = 2.4,	M = 2.6,	M = 2.6,	M = 2.5,	
		ME = 2.4,	ME = 2.6,	ME = 2.7,	ME = 2.4,	
		SD = 0.2,	SD = 0.2,	SD = 0.4,	SD = 0.4,	
		Min = 2.0,	Min = 2.3,	Min = 1.8,	Min = 2.0,	
		Max = 2.6	Max = 2.9	Max = 3.1	Max = 3.2	
Phosphorus (mmol/L)	0.7–1.6 ^3^	*N* = 10,	*N* = 10,	*N* = 10,	*N* = 10,	χ^2^ = 25.973, *df* = 3, *p* < 0.001
		M = 1.5,	M = 1.6,	M = 2.7,	M = 4.1,	
		ME = 1.6,	ME = 1.3,	ME = 2.3,	ME = 3.8,	
		SD = 0.5,	SD = 0.6,	SD = 1.0,	SD = 1.2,	
		Min = 0.8,	Min = 1.1,	Min = 1.9,	Min = 2.9,	
		Max = 2.2	Max = 2.6	Max = 4.7	Max = 6.5	
Potassium (mmol/L)	3.5–5.1 ^3^	*N* = 10,	*N* = 10,	*N* = 10,	*N* = 10,	χ^2^ = 11.392, *df* = 3, *p* = 0.010
		M = 4.5,	M = 5.2,	M = 5.4,	M = 5.1,	
		ME = 4.5,	ME = 5.2,	ME = 5.4,	ME = 4.8,	
		SD = 0.3,	SD = 0.5,	SD = 1.3,	SD = 0.9,	
		Min = 4.2,	Min = 4.5,	Min = 2.9,	Min = 3.7,	
		Max = 4.9	Max = 6.2	Max = 7.3	Max = 6.8	
Sodium (mmol/L)	140.0–155.0 ^3^	*N* = 10,	*N* = 10,	*N* = 10,	*N* = 10,	χ^2^ = 0.962, *df* = 3, *p* = 0.810
		M = 147.1,	M = 147.9,	M = 146.4,	M = 148.5,	
		ME = 147.0,	ME = 148.5,	ME = 148.0,	ME = 149.5,	
		SD = 4.9,	SD = 3.7,	SD = 8.9,	SD = 4.7,	
		Min = 141.0,	Min = 139.0,	Min = 128.0,	Min = 137.0,	
		Max = 156.0	Max = 153.0	Max = 157.0	Max = 154.0	
Magnesium (mmol/L)	0.6–1.3 ^3^	*N* = 10,	*N* = 8,	*N* = 10,	*N* = 9,	χ^2^ = 16.589, *df* = 3, *p* < 0.001
		M = 0.8,	M = 1.0,	M = 1.2,	M = 1.1,	
		ME = 0.8,	ME = 1.0,	ME = 1.1,	ME = 1.1,	
		SD = 0.1,	SD = 0.1,	SD = 0.4,	SD = 0.2,	
		Min = 0.7,	Min = 0.9,	Min = 0.7,	Min = 0.9,	
		Max = 1.0	Max = 1.2	Max = 1.8	Max = 1.4	
Total protein (g/L)	54.0–75.0 ^3^	*N* = 10,	*N* = 10,	*N* = 10,	*N* = 10,	χ^2^ = 3.312, *df* = 3, *p* = 0.346
		M = 55.3,	M = 62.3,	M = 59.8,	M = 62.8,	
		ME = 57.3,	ME = 63.9,	ME = 63.5,	ME = 58.8,	
		SD = 7.9,	SD = 9.3,	SD = 11.9,	SD = 15.0,	
		Min = 43.1,	Min = 41.4,	Min = 40.5,	Min = 50.7,	
		Max = 67.9	Max = 73.6	Max = 72.7	Max = 102.1	
Albumin (g/L)	25.0–44.0 ^3^	*N* = 10,	*N* = 10,	*N* = 10,	*N* = 10,	χ^2^ = 4.453, *df* = 3, *p* = 0.217
		M = 36.0,	M = 36.6,	M = 31.9,	M = 30.9,	
		ME = 37.3,	ME = 36.6,	ME = 31.8,	ME = 31.9,	
		SD = 7.0,	SD = 7.5,	SD = 10.3,	SD = 5.1,	
		Min = 28.1,	Min = 23.5,	Min = 13.0,	Min = 23.1,	
		Max = 45.0	Max = 46.7	Max = 43.2	Max = 38.2	
Globulin (g/L)	<45.0 ^3^	*N* = 10,	*N* = 10,	*N* = 10,	*N* = 10,	χ^2^ = 10.342, *df* = 3, *p* = 0.016
		M = 19.3,	M = 25.7,	M = 27.9,	M = 31.9,	
		ME = 17.9,	ME = 27.0,	ME = 27.6,	ME = 28.9,	
		SD = 5.0,	SD = 4.6,	SD = 7.2,	SD = 15.5,	
		Min = 15.0,	Min = 17.9,	Min = 16.3,	Min = 14.4,	
		Max = 31.9	Max = 31.2	Max = 42.2	Max = 72.2	
Glucose (mmol/L)	3.05–6.1 ^3^	*N* = 10,	*N* = 9,	*N* = 10,	*N* = 10,	χ^2^ = 2.049, *df* = 3, *p* = 0.562
		M = 5.6,	M = 7.7,	M = 9.2,	M = 5.2,	
		ME = 5.7,	ME = 5.1,	ME = 5.5,	ME = 5.3,	
		SD = 1.7,	SD = 9.0,	SD = 11.5,	SD = 0.8,	
		Min = 2.5,	Min = 3.6,	Min = 3.7,	Min = 3.9,	
		Max = 7.8	Max = 31.6	Max = 41.9	Max = 6.8	
Fructosamine (mmol/L)	<374.0 ^3^	*N* = 10,	*N* = 10,	*N* = 10,	*N* = 10,	χ^2^ = 5.510, *df* = 3, *p* = 0.138
		M = 272.1,	M = 346.2,	M = 285.4,	M = 269.1,	
		ME = 274.0,	ME = 305.8,	ME = 273.0,	ME = 262.6,	
		SD = 59.7,	SD = 127.0,	SD = 109.0,	SD = 45.9,	
		Min = 193.8,	Min = 187.5,	Min = 160.2,	Min = 213.9,	
		Max = 369.5	Max = 661.6	Max = 556.9	Max = 363.1	
Alpha-amylase (U/L)	<1650.0 ^3^	*N* = 10,	*N* = 8,	*N* = 10,	*N* = 9,	χ^2^ = 14.052, *df* = 3, *p* = 0.003
		M = 601.7,	M = 1059.4,	M = 1565.7,	M = 1979.1,	
		ME = 481.5,	ME = 926.0,	ME = 1680.5,	ME = 1597.0,	
		SD = 271.5,	SD = 491.7,	SD = 574.1,	SD = 2003.5,	
		Min = 305.0,	Min = 452.0,	Min = 600.0,	Min = 8.0,	
		Max = 1177.0	Max = 1938.0	Max = 2313.0	Max = 7053.0	
DGGR-lipase (U/L)	<120.0 ^3^	*N* = 10,	*N* = 10,	*N* = 10,	*N* = 10,	χ^2^ = 16.500, *df* = 3, *p* < 0.001
		M = 55.8,	M = 111.6,	M = 237.0,	M = 821.5,	
		ME = 48.3,	ME = 103.4,	ME = 223.0,	ME = 178.0,	
		SD = 28.1,	SD = 45.0,	SD = 132.6,	SD = 1317.4,	
		Min = 14.1,	Min = 62.8,	Min = 30.3,	Min = 23.4,	
		Max = 104.5	Max = 193.7	Max = 486.2	Max = 3316.2	
Bilirubin (µmol/L)	<3.4 ^3^	*N* = 10,	*N* = 10,	*N* = 9,	*N* = 10,	χ^2^ = 11.793, *df* = 3, *p* = 0.008
		M = 1.0,	M = 1.1,	M = 2.4,	M = 2.1,	
		ME = 1.0,	ME = 1.1,	ME = 1.3,	ME = 2.2,	
		SD = 0.6,	SD = 0.6,	SD = 1.8,	SD = 0.6,	
		Min = 0.1,	Min = 0.2,	Min = 0.4,	Min = 1.3,	
		Max = 1.9	Max = 1.9	Max = 5.2	Max = 2.8	
Alanine aminotransferase (U/L)	<88.0 ^3^	*N* = 10,	*N* = 10,	*N* = 10,	*N* = 10,	χ^2^ = 2.571, *df* = 3, *p* = 0.463
		M = 45.1,	M = 73.6,	M = 444.1,	M = 63.9,	
		ME = 42.8,	ME = 61.9,	ME = 70.8,	ME = 60.6,	
		SD = 21.5,	SD = 59.2,	SD = 1059.7,	SD = 26.8,	
		Min = 17.8,	Min = 20.9,	Min = 11.3,	Min = 24.3,	
		Max = 81.9	Max = 213.7	Max = 3433.1	Max = 123.4	
Alkaline phosphatase (U/L)	<147.0 ^3^	*N* = 10,	*N* = 10,	*N* = 10,	*N* = 10,	χ^2^ = 6.151, *df* = 3, *p* = 0.104
		M = 91.7,	M = 72.0,	M = 479,9,	M = 53.5,	
		ME = 59.5,	ME = 45.0,	ME = 148.0,	ME = 50.5,	
		SD = 73.0,	SD = 92.9,	SD = 737.0,	SD = 31.4,	
		Min = 14.0,	Min = 19.0,	Min = 10.0,	Min = 20.0,	
		Max = 216.0	Max = 331.0	Max = 2336.0	Max = 104.0	
Aspartate aminotransferase (U/L)	<51.0 ^3^	*N* = 10,	*N* = 10,	*N* = 10,	*N* = 10,	χ^2^ = 2.210, *df* = 3, *p* = 0.530
		M = 32.1,	M = 44.5,	M = 50.6,	M = 41.7,	
		ME = 30.8,	ME = 34.5,	ME = 39.3,	ME = 40.0,	
		SD = 13.2,	SD = 33.4,	SD = 33.9,	SD = 20.3,	
		Min = 12.7,	Min = 23.4,	Min = 16.9,	Min = 20.0,	
		Max = 61.6	Max = 133.6	Max = 121.8	Max = 79.6	
Glutamate dehydrogenase (U/L)	<8.0 ^3^	*N* = 10,	*N* = 10,	*N* = 10,	*N* = 10,	χ^2^ = 8.178, *df* = 3, *p* = 0.042
		M = 7.7,	M = 15.3,	M = 62.7,	M = 3.5,	
		ME = 4.9,	ME = 7.1,	ME = 16.9,	ME = 3.8,	
		SD = 6.7,	SD = 21.2,	SD = 112.8,	SD = 2.51,	
		Min = 1.7,	Min = 2.7,	Min = 1.8,	Min = 0.1,	
		Max = 23.3	Max = 70.5	Max = 350.2	Max = 6.6	
Creatine kinase (U/L)	<200.0 ^3^	*N* = 10,	*N* = 10,	*N* = 10,	*N* = 10,	χ^2^ = 1.507, *df* = 3, *p* = 0.681
		M = 208.8,	M = 188.0,	M = 179.8,	M = 234.0,	
		ME = 141.5,	ME = 143.0,	ME = 136.0,	ME = 164.5,	
		SD = 238.0,	SD = 140.6,	SD = 94.4,	SD = 168.9,	
		Min = 53.0,	Min = 51.0,	Min = 54.0,	Min = 93.0,	
		Max = 857.0	Max = 515.0	Max = 329.0	Max = 528.0	
Triglycerides (mmol/L)	<3.9 ^3^	*N* = 10,	*N* = 8,	*N* = 10,	*N* = 9,	χ^2^ = 9.269, *df* = 3, *p* = 0.026
		M = 0.7,	M = 1.5,	M = 1.2,	M = 0,9,	
		ME = 0.6,	ME = 0.9,	ME = 0,9,	ME = 0.8,	
		SD = 0.3,	SD = 1.6,	SD = 0.6,	SD = 0.6,	
		Min = 0.3,	Min = 0.8,	Min = 0.6,	Min = 0.3,	
		Max = 1.4	Max = 5.5	Max = 2.3	Max = 2.2	

FGF-23 = Fibroblast growth factor-23; RI = reference interval ^1^ Statistical differences between study groups I–IV, Kruskal–Wallis test with Bonferroni correction, *p* < 0.05 stated as statistically significant; ^2^ reference range is based on [28]; ^3^ reference ranges were based on the internal values of the LABOKLIN laboratory (Bad Kissingen, Germany); ^A^ FGF-23 ELISA Kit, Kainos Laboratories, Tokyo, Japan; ^B^ COBAS 8000, Roche Diagnostics, Germany.

**Table 3 animals-13-03202-t003:** Correlation between biochemical parameters and fibroblast growth factor (FGF)-23 using Spearman’s rank correlation coefficient.

		FGF-23 ^A^	Crea	Urea	P	K	Mg	Ca	Alpha-Amyl.	DGGR Lipase	Bil	Trigly	Glu	Fra	TP	Alb	Glob
FGF-23 ^A^	CC	1.000	0.899 **	0.831 **	0.734 **	0.305	0.521 **	0.258	0.548 **	0.518 **	0.458 **	0.129	−0.153	−0.193	0.109	−0.282	0.418 **
	Sig	-	<0.001	<0.001	<0.001	0.055	<0.001	0.108	<0.001	<0.001	0.003	0.446	0.352	0.234	0.505	0.078	0.007
	*N*	40	40	40	40	40	37	40	37	40	39	37	39	40	40	40	40
Crea	CC	0.899 **	1.000	0.925 **	0.788 **	0.257	0.582 **	0.600 **	0.535 **	0.600 **	0.552 **	0.119	−0.050	−0.133	0.078	−0.331 *	0.440 **
	Sig	<0.001	-	<0.001	<0.001	0.110	<0.001	<0.001	<0.001	<0.001	<0.001	0.481	0.760	0.412	0.634	0.037	0.005
	*N*	40	40	40	40	40	37	40	37	40	39	37	39	40	40	40	40
Urea	CC	0.831 **	0.925 **	1.000	0.809 **	0.269	0.657 **	0.585 **	0.532 **	0.585 **	0.482 **	0.066	−0.065	−0.192	0.074	−0.301	0.364 *
	Sig	<0.001	<0.001	-	<0.001	0.093	<0.001	<0.001	<0.001	<0.001	0.002	0.698	0.692	0.234	0.652	0.059	0.021
	*N*	40	40	40	40	40	37	40	37	40	39	37	39	40	40	40	40
P	CC	0.734 **	0.788 **	0.809 **	1.000	0.203	0.417 *	0.363 *	0.437 **	0.363 *	0.470 **	−0.066	0.054	−0.332 *	−0.070	0.497 **	0.392 *
	Sig	<0.001	<0.001	<0.001	-	0.209	0.010	0.021	0.007	0.021	0.003	0.696	0.746	0.036	0.666	0.001	0.012
	*N*	40	40	40	40	40	37	40	37	40	39	37	39	40	40	40	40
K	CC	0.305	0.257	0.269	0.203	1.000	0.528 **	0.264	0.344 *	0.264	−0.139	0.371 *	−0.199	0.109	0.244	0.028	0.312 *
	Sig	0.055	0.110	0.093	0.209	-	<0.001	0.099	0.037	0.099	0.398	0.024	0.226	0.505	0.129	0.866	0.050
	*N*	40	40	40	40	40	37	40	37	40	39	37	39	40	40	40	40
Mg	CC	0.521 **	0.582 **	0.657 **	0.417 *	0.528 **	1.000	0.639 **	0.327 *	0.639 **	0.332 *	0.205	−0.063	0.188	0.398 *	−0.024	0.450 **
	Sig	<0.001	<0.001	<0.001	0.010	<0.001	-	<0.001	0.049	<0.001	0.048	0.223	0.713	0.264	0.015	0.890	0.005
	*N*	37	37	37	37	37	40	37	37	37	36	37	36	37	37	37	37
Ca	CC	0.258	0.600 **	0.585 **	0.363 *	0.264	0.639 **	1.000	−0.042	0.313 *	0.340 *	0.002	−0.138	0.490 **	0.490 **	0.510 **	0.116
	Sig	0.108	<0.001	<0.001	0.021	0.099	<0.001	-	0.806	0.049	0.034	0.992	0.402	0.001	0.001	<0.001	0.476
	*N*	40	40	40	40	40	37	40	37	40	39	37	36	40	40	40	40
Alpha-amyl.	CC	0.548 **	0.535 **	0.532 **	0.437 **	0.344 *	0.327 *	−0.042	1.000	0.445 **	0.184	0.202	0.097	−0.418 *	−0.214	0.492 **	0.286
	Sig	<0.001	<0.001	<0.001	0.007	0.037	0.049	0.806	-	0.006	0.283	0.230	0.573	0.010	0.203	0.002	0.087
	*N*	37	37	37	37	37	37	37	40	37	36	37	36	37	37	37	37
DGGR lipase	CC	0.518 **	0.600 **	0.585 **	0.363 *	0.264	0.639 **	0.313 *	0.445 **	1.000	0.340 *	0.002	−0.042	0.101	0.158	0.043	0.200
	Sig	<0.001	<0.001	<0.001	0.021	0.099	<0.001	0.049	0.006	-	0.034	0.992	0.798	0.536	0.330	0.795	0.216
	*N*	40	40	40	40	40	37	40	37	40	39	37	39	40	40	40	40
Bil	CC	0.458 **	0.552 **	0.482 **	0.470 **	−0.139	0.332 *	0.340 *	0.184	0.340 *	1.000	0.007	0.315	0.046	0.187	−0.094	0.278
	Sig	0.003	<0.001	0.002	0.003	0.398	0.048	0.034	0.283	0.034	-	0.965	0.054	0.779	0.255	0.567	0.086
	*N*	39	39	39	39	39	36	39	36	39	40	36	38	39	39	39	39
GLDH	CC	−0.131	−0.157	−0.072	−0.037	0.062	0.181	0.277	−0.042	0.277	0.049	0.008	0.138	0.292	0.300	0.262	0.141
	Sig	0.419	0.333	0.657	0.821	0.706	0.283	0.084	0.804	0.084	0.767	0.962	0.404	0.067	0.060	0.102	0.385
	*N*	40	40	40	40	40	37	40	37	40	39	37	39	40	40	40	40
Trigly	CC	0.129	0.119	0.066	−0.066	0.371 *	0.205	0.002	0.202	0.002	0.007	1.000	−0.175	0.185	0.192	0.008	0.251
	Sig	0.446	0.481	0.698	0.696	0.024	0.223	0.992	0.230	0.992	0.965	-	0.306	0.274	0.255	0.964	0.134
	*N*	37	37	37	37	37	37	37	37	37	36	40	36	37	37	37	37
Glu	CC	−0.153	−0.050	−0.065	0.054	−0.199	−0.063	−0.138	0.097	−0.042	0.315	−0.175	1.000	−0.085	−0.188	−0.251	−0.103
	Sig	0.352	0.760	0.692	0.746	0.226	0.713	0.402	0.573	0.798	0.054	0.306	-	0.609	0.251	0.123	0.532
	*N*	39	39	39	39	39	36	36	36	39	38	36	40	39	39	39	39
Fra	CC	−0.193	−0.133	−0.192	−0.332 *	0.109	0.188	0.490 **	−0.418 *	0.101	0.046	0.185	−0.085	1.000	0.701 **	0.705 **	0.174
	Sig	0.234	0.412	0.234	0.036	0.505	0.264	0.001	0.010	0.536	0.779	0.274	0.609	-	<0.001	<0.001	0.283
	*N*	40	40	40	40	40	37		37	40	39	37	39	40	40	40	40
TP	CC	0.109	0.078	0.074	−0.070	0.244	0.398 *	0.490 **	−0.214	0.158	0.187	0.192	−0.188	0.701 **	1.000	0.612 **	0.588 **
	Sig	0.505	0.634	0.652	0.666	0.129	0.015	0.001	0.203	0.330	0.255	0.255	0.251	<0.001	-	<0.001	<0.001
	*N*	40	40	40	40	40	37	40	37	40	39	37	39	40	40	40	40
Alb	CC	−0.282	−0.331 *	−0.301	−0.497 **	0.028	−0.024	0.510 **	0.492 **	0.043	−0.094	0.008	−0.251	0.705 **	0.612 **	1.000	−0.252
	Sig	0.078	0.037	0.059	0.001	0.866	0.890	<0.001	0.002	0.795	0.567	0.964	0.123	<0.001	<0.001	-	0.116
	*N*	40	40	40	40	40	37	40	37	40	39	37	39	40	40	-	40
Glob	CC	0.418 **	0.440 **	0.364 *	0.392 *	0.312 *	0.450 **	0.116	0.286	0.200	0.278	0.251	−0.103	0.174	0.558 **	−0.252	1.000
	Sig	0.007	0.005	0.021	0.012	0.050	0.005	0.476	0.087	0.216	0.086	0.134	0.557	0.283	<0.001	0.116	-
	*N*	40	40	40	40	40	37	40	37	40	39	37	35	40	40	40	40

Alb = albumin; Alpha-amyl. = alpha-amylase; Bil = bilirubin; Ca = calcium; CC = correlation coefficient; Crea = creatinine; FGF-23 = fibroblast growth factor-23; Fra = Fructosamine; GLDH = glutamate dehydrogenase; Glob = globulin; Glu = glucose; Mg = magnesium; Na = sodium; K = potassium; P = phosphorus; Sig = statistical significance; TP = total protein; Trigly = triglycerides; ^A^ FGF-23 ELISA Kit, Kainos Laboratories, Tokyo, Japan; * Correlation is significant at the 0.01 level (2-tailed); ** correlation is significant at the 0.05 level (2-tailed).

## Data Availability

Not applicable.

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
