# Peer review of "Fibroblast Growth Factor-23 (FGF-23) in Dogs—Reference Interval and Correlation with Hematological and Biochemical Parameters"

_animals, 2023, doi:10.3390/ani13203202_

Round 1
Reviewer 1 Report
In the first part, the authors present an observational study on the variation in the serum concentration of FGF-23 in clinically stable dogs. Possibly the greatest contribution of this study is the demonstration that the serum concentration of FGF-23 presented a wide reference interval in clinically stable dogs. Therefore, the interpretation of the FGF-23 result must be performed together with the other parameters of the renal profile. In the first part of the present study, FGF-23 was evaluated in 136 clinically healthy dogs, and the performance of the kit was considered good, because the coefficient of variation was low (<15%) for both intra- and interassay precision.
The authors evaluated the FGF-23 concentration in dogs using the Kainos ELISA FGF-23 kit. This kit has already been used in other studies for the analysis of FGF-23 in dogs and cats (control group versus chronic kidney disease group).In these previous studies, FGF-23 concentrations increased with severity of renal azotemia, and FGF‐23 concentrations were positively correlated with creatinine and phosphorus concentrations. Similar results were observed by the authors; therefore, the originality of the study does not revolve around the use of the Kainos ELISA FGF-23 kit or the clinical condition of the dogs.The interpretation of the FGF-23 result must be performed together with the other parameters of the renal profile, and this issue has not been deepened in previous studies.
In the first part of the study, the authors determined the serum FGF-23 concentration in dogs with hematological and biochemical results according to the reference intervals of the laboratory. Apparently, the authors did not have access to other data such as regarding history, clinical signs, urinalysis, or diagnostic imaging of all animals, including those older than 9 years.
The serum concentration of FGF-23 was higher in the group of dogs aged between 9 and 16 years compared to the group of dogs aged between 1 and 8 years. In dogs of ≥ 9 years of age FGF-23 ranged from 128.8 pg/ml to 635.0 pg/ml. This result was not very different from groups I and II (second part of the study). Furthermore, the authors did not report the creatinine values of dogs > 9 years of age. Are all dogs between the ages of 9 and 16 considered clinically healthy?
To clarify this issue, it would be good to evaluate the concentration of FGF-23 in another group of dogs aged over 9 years, but in this case, considering a larger set of information, such as regarding history, clinical signs, urinalysis, and diagnostic imaging.
In the second part of the present study, FGF-23 was evaluated in dogs with or without azotemia, and the highest concentrations of FGF-23 were observed in dogs with creatinine above 265 umol/l. In addition, a positive correlation was observed between the serum concentration of FGF-23 and renal biomarkers such as creatinine, urea and phosphorus.
The present study draws attention to the methodology for analyzing FGF-23 using the Kainos ELISA kit, such as the need for diagnostic cutoffs and the wide reference range.
The correlation between FGF-23 concentration and hematological and other biochemical parameters was weak, although statistically significant.
The calculation of reference values for FGF-23 concentration is useful, as this parameter can contribute to the evaluation and follow-up of dogs with renal azotemia.
However, the interpretation of the FGF-23 result must be performed together with the other parameters of the renal profile, considering a wide reference interval and similar results between clinically healthy dogs and dogs in the early stages of renal disease.
Table 3 is not clear. The current format of this table is making it difficult to understand the results of the correlation analysis.
Reviewer 2 Report
Line 40: write also FGF-23 as keyword
Line 50: "blood pressure measurement" instead of "measurements"
Line 82: these group of patients cannot be considered as CKD dogs; furthermore it's written also in the discussion section that between them there can be healthy as well as AKI patients with azotemia due to different causes. If this group is better to be considered in the manuscript, other information are needed in order to have a CKD diagnosis (e.g. presence/absence of proteinuria, urine specific gravity, evaluation of kidney morphology) as well as current or prior medications known to influence phosphorus/PTH concentrations
Line 94: there is a closing parenthesis at the end of the sentence
Line 99: add bibliographic citation to justify time and temperature storing of FGF-23. Even a centrifugation after 8 hours from the venipuncture can influence FGF-23 concentrations as written in the article by Smith E.R, et al (E. R. Smith, L. P. McMahon, and S. G. Holt, “Method-specific differences in plasma fibroblast growth factor 23 measurement using four commercial ELISAs,” Clinical Chemistry and Laboratory Medicine, vol. 51, no. 10, pp. 1971–1981, 2013)
Line 106: has it been evaluated prior or current medications that can influence phosphorus concentrations? (e.g. phosphate binders)
Line 107: specify that this is a kit used to determine human FGF-23
Line 116: were the samples run in duplicate?
Line 125: as written in line 99 it's better to add a bibliographic citation to justify time and temperature of FGF-23 storing
Line 130: as written in the introduction section (line 82), these patients have no background information so there cannot be staged with IRIS guidelines staging of CKD
Line 172: figure 1. The figure and the caption are not centered
Line 183: in the figure 3 it will be helpful to add graphically the P value between study groups
Linea 189: are you referring to the patients included as CKD? It is not clear written like this
Line 201: as for figure 3 it will be helpful to add graphically the P value between study groups
Line 216: Table 1 it's better to be considered as supplemental material focusing on Table 2. Also there should be more space between parameters to better see all the results
Line 246: the table is not understandable; the parameters are all too close to each other
Line 290: it cannot be written in the study aim that CKD patients have been included; perhaps they are not even CKD patients, as you write in this section
Line 297: it cannot be written that FGF-23 is of limited value in CKD early diagnosis when the study group consisted of patients with multiple causes for azotemia
Line 300: this statement should be together with the one written in line 290
Line 305: justify the sentence adding citations to explain postprandial influence on FGF-23 concentration. As an example, in this human manuscript it is highlighted FGF-23 variations after 12 hour from a meal (E. R. Smith, M. M. Cai, L. P. McMahon, and S. G. Holt, “Biological variability of plasma intact and C-terminal FGF23 measurements,” The Journal of Clinical Endocrinology and Metabolism, vol. 97, no. 9, pp. 3357–3365, 2012)
Line 317: you cannot define a weak correlation between FGF-23 and serum phosphorus concentration when you do not know if patients were treated to drugs known to influence phosphorus concentrations (e.g., renal diet, phosphate binders)
Line 339: it would be better to end the sentence by adding that prospective studies with a wider range of patients are needed in order to understand better the FGF-23 - Magnesium correlation
Line 356: as written in line 339 prospective studies are needed to better understand FGF-23 and reticulocyte hemoglobin concentration correlation
Reviewer 3 Report
The reviewer thanks that authors for a comprehensive manuscript creating a reference interval and attempt at correlation of various biomarkers ( hematological and biochemical parameters) for an FGF-23 assay developed at their place of employment. The study provides information from which larger studies may be developed utilizing the utility of FGF-23 assays for monitoring progressive and perhaps even response to therapeutic management of CKD in dogs. Prospective studies will need to be performed.The major limitation seems to be that the samples assessed came from samples submitted to the laboratory without much clinical data available that could be obtained with a prospective study.
In Materials and Methods
Healthy dogs have no urinalysis submitted. How then is the determination of health status of the kidneys is determined for these dogs without a USG minimally?
Lines 153 -160: Population of clinically healthy dogs is provided but no weights are listed. Is this information available from sample submission or at least calling the hospital sending in the sample.
Firgure 2: It may have been the way that this figure printed out but did not provide much information. Would consider deleting it. Is the box what you are calling the mean/median ? The entire graph does not appear to be a typical whiskers diagram.
There is a bit of confusion in this reviewer's mind about the creation of the study group I-IV that looked at creatinine and then presentation of the results for these 4 groups presenting their FGF-23 and their creatinine levels. How is the ready to know if these animals with say group 4 creatinine levels were they all in range of a creatinine > 5 mg/dl with defined CKD or might have their been AKIs included. Please provide justification as to why this approach was utilized.
What was the time line for data collection? Perhaps I missed this in my review. And all data parameters were collected on the same analyzer throughout the study period? There is quite of information presented in TABLES 1 and 2 that make them very difficult to read and actually determine those that are significant. Perhaps breaking the information down into smaller tables . example one for Hematology and the other for biochemistry data. Table 1 and Table 2 seem to present the same data sets, are they both necessary.
The formatting of Table 3 makes it essentially indecipherable. Way too much information there is a table form without a divisional grid.
The discussion is concise and does bring up some of the limitations. The greatest being that samples were collected from left over blood from submissions to the laboratory from outside hospitals and relied upon the information provided by the individual hospitals. Additionally, the blood left over from tests submitted, did the authors confirm that the blood work was not retest from the same patient so duplicate case enrolled with blood obtained from same patient at different times. Was this taken into consideration in terms of study design or was duplication of enrolled patients reason for exclusion?
Line 363 in Discussion: Typographical error. There are 2 "INs" at the start of this line. Delete one of the "INs"
This is a difficult review in that the first part of the manuscript essentially describes validation of the assay for FGF -23 in canines. Something that is much needed. This portion is worthy of publication.
The second portion where attempt to correlate the FGF-23 data obtained with hematological and biochemical parameters is challenged by the fact that this was left over blood and the reason for submission is unclear. The authors create groups from dogs without the knowledge of whether they have acute or chronic kidney dysfunction while evaluating an assay that at its best may have more utilization predictive of progression of CKD. Although this format has been use by others validating assays for biomarkers , the need is there to obtain more background information about the dogs that prompted the submitting veterinarian to perform a CBC and chemistry profile. A larger case enrollment would also be recommended
Round 2
Reviewer 2 Report
Line 82: it would be better write "following IRIS guidelines"
Line 193: it would be better end the sentence with "clinical chemistry" and begin another one with "Forty dogs..."
Line 231: it would be better add a space line between the caption of Table 1 and the results
Line 257: Table 3. In K line the numbers are written in another font type, as well as in Mg line
Line 303: the sentence should be ended that, for these reasons, this is a limit of this study (including azotemic patients without having background information about the azotemia origin)
